# Ethnically biased microsatellites contribute to differential gene expression and glutathione metabolism in Africans and Europeans

Nick Kinney[1,2¤]*, Lin Kang[1,2], Harpal Bains[1], Elizabeth Lawson[1], Mesam Husain[1], Kumayl Husain[1], Inderjit Sandhu[1], Yongdeok Shin[1], Javan K. Carter[3], Ramu Anandakrishnan[1,2], Pawel Michalak[1,2,4], Harold Garner[1,2]

1 Edward Via College of Osteopathic Medicine, Blacksburg, Virginia, United States of America, 2 Gibbs Cancer Center & Research Institute, Spartanburg, South Carolina, United States of America, 3 University of Colorado Boulder, Boulder, Colorado, United States of America, 4 Institute of Evolution, University of Haifa, Haifa, Israel

¤ Current address: Dr. Nicholas Kinney, Center for Bioinformatics and Genetics, Edward Via College of Osteopathic Medicine, Blacksburg, Virginia, United States of America
* nkinney06@gmail.com

**Data Availability Statement:** All relevant data are within the paper and its Supporting information files.

## Abstract

Approximately three percent of the human genome is occupied by microsatellites: a type of short tandem repeat (STR). Microsatellites have well established effects on (a) the genetic structure of diverse human populations and (b) expression of nearby genes. These lines of inquiry have uncovered 3,984 ethnically biased microsatellite loci (EBML) and 28,375 expression STRs (eSTRs), respectively. We hypothesize that a combination of EBML, eSTRs, and gene expression data (RNA-seq) can be used to show that microsatellites contribute to differential gene expression and phenotype in human populations. In fact, our previous study demonstrated a degree of mutual overlap between EBML and eSTRs but fell short of quantifying effects on gene expression. The present work aims to narrow the gap. First, we identify 313 overlapping EBML/eSTRs and recapitulate their mutual overlap. The 313 EBML/eSTRs are then characterized across ethnicity and tissue type. We use RNA-seq data to pursue validation of 49 regions that affect whole blood gene expression; 32 out of 54 affected genes are differentially expressed in Africans and Europeans. We quantify the relative contribution of these 32 genes to differential expression; fold change tends to be less than other differentially expressed genes. Repeat length correlates with expression for 15 of the 32 genes; two are conspicuously involved in glutathione metabolism. Finally, we repurpose a mathematical model of glutathione metabolism to investigate how a single polymorphic microsatellite affects phenotype. We conclude with a testable prediction that microsatellite polymorphisms affect GPX7 expression and oxidative stress in Africans and Europeans.

## Introduction

The human genome contains an abundance of repetitive DNA; on the order of five million regions spanning 1.5 billion base pairs [1]. These regions are broadly classified by the size and

**Funding:** The authors received no specific funding for this work.

distribution of their repeat motif. Short and long interspersed repeats–SINEs and LINEs–are spread throughout the genome. On the contrary, short and long tandem repeats–microsatellites and minisatellites–are arranged side-by-side. In the case of microsatellites, each side-by-side arrangement consists of a one to six base pair motif copied up to 100 times. Over 600,000 microsatellites exist in the human genome [2,3], with approximately 50,000 embedded in gene introns, exons, and untranslated regions (UTRs). Microsatellites have been used for decades as putatively neutral markers in forensic and population analysis; however, their potential function has long been debated [2].

Insight into microsatellite function began with the availability of genome sequences. Early studies observed an abundance of various trinucleotide microsatellites in intergenic regions of vertebrates and various dinucleotide microsatellites at recombination hot spots [4,5]. In humans, microsatellites have been associated with promoters and gene start sites [6]. In general, the distribution of microsatellites is not explained by chance alone, yet their function (if any) is hard to pin down [7].

The function of individual microsatellites probably stems from a complex combination of their repeat unit, length, and location in the genome. Disease causing tri-nucleotide repeats are de facto examples. Machado-Joseph disease, Huntington's disease, and various ataxias are all caused by elongated poly-glutamine (CAG) repeats [8]. However, these dramatic effects are rare. Recent studies show that most microsatellites have subtle effects on gene expression. In 2015, a breakthrough study used variance partitioning to quantify these subtle effects [9]. A correlation between repeat length and nearby gene expression was established for 2,060 microsatellites: subsequently called expression short tandem repeats (eSTRs). In 2019, the same approach was scaled up and applied to data from the Genotype-Tissue Expression Project [10]. The unprecedented scale of this this study revealed 28,375 eSTRs associated with 12,494 genes across 17 tissue types [10].

Older studies seeking to link gene expression with nearby microsatellite repeat length were relegated to candidate gene studies. A 2002 study investigated effects of a penta-nucleotide microsatellite in the promoter binding site of PIG3 [11]: an enzyme involved in oxidative stress. Longer repeats resulted in more efficient transcriptional activation through p53 promoter binding [11]. This association was reproduced in the aforementioned 2015 study using variance partitioning [9]. On the other hand, dinucleotide repeats associated with MMP9 and EGFR were not [9]. These two associations were not reproduced until variance partitioning was scaled up in 2019 [10]. These observations have two implications. First, there is a need for high-powered studies and methods to detect undiscovered microsatellite polymorphisms associated with gene expression. Second, there is a need to reproduce (or refute) known associations.

Scientific interest in microsatellites is not limited to their effects on gene expression. The high mutation rate of microsatellites has been leveraged to study the structure of diverse human populations for decades. A groundbreaking 1994 study collected genetic data for 30 polymorphic microsatellites in 148 individuals from 14 human populations [12]. A neighbor joining tree of these individuals was remarkably consistent with their geographic origin: Africa, Oceania, East Asia, Europe, and America [12]. This finding was a departure from mitochondrial based trees and set a precedent for future studies of microsatellite diversity. Indeed, studies in 1997 and 2003 scaled up analysis to 60 and 377 microsatellite loci, respectively [13,14]. Both reiterated abundant microsatellite diversity and private alleles within African populations. This body of research culminated with a 2009 study of 121 African populations, four African American populations, and 60 non-African populations [15]. Genetic diversity was shown to decline with distance from Africa and private alleles were more numerous in Africa than other regions [15].

Relatively few studies have sought to link these two lines of research; specifically, how polymorphic microsatellites contribute to differential gene expression in human populations. The links that have been made are limited to microsatellites embedded in specific genes such as tyrosine hydroxylase (TH) and carnosinase dipeptidase (CNDP1). In the case of TH, a perfect allele–rare in Caucasians–is associated with genetic predisposition to schizophrenia [16]. In the case of *CNDP1*, the Mannheim allele is associated with a decreased risk for type 2 renal disease in European Americans [17]. In fact, a bold perspective was published prior to both findings; it was proposed that microsatellites supply abundant mutations that have quantitative effects on phenotype [18]. A follow-up review by the same authors cited evidence that microsatellites are leveraged by natural selection in several model organisms including humans [19]. Several emerging questions were highlighted. How is phenotypic variation related to repeat length in corresponding microsatellites? To what extent do microsatellites contribute to divergence in natural populations?

Recent inroads were made by our own efforts to characterize germline microsatellites variations in the 1000 Genomes Project. First, we established a database of microsatellite variation across population, ethnicity (super population) and gender [20]. On the heels of this effort, we identified 3,984 ethnically biased microsatellite loci (EBML) in Africans, South Asians, East Asians, Europeans, and Americans [21]. Our main findings showed that EBML and eSTRs possess mutual overlap suggesting that microsatellites contribute to differential gene expression in human populations [21]. However, this possibility was not rigorously validated with gene expression data. Furthermore, the association was established prior to the discovery of 28,375 eSTRs in 2019 [10]. We hypothesize that a combination of EBML, newly discovered eSTRs, and gene expression data (RNA-seq) can be used to show that microsatellites contribute to differential gene expression and phenotype in human populations.

The test of our hypothesis unfolds in 5 sections. First, we characterize the overlap between EBML and eSTRs across ethnicity and tissue type; in the case of whole blood, we identify 49 overlapping EBML/eSTRs associated with 54 genes. Second, we use RNA-seq data to show that 32 of these genes are differentially expressed in Africans and Europeans. Third, we quantify the relative contribution of the 32 genes to differential expression; fold change for these genes tends to be less than all other differentially expressed genes. Fourth, we use linear regression to show that repeat length correlates with expression for 15 of the 32 genes: two are conspicuously involved in glutathione metabolism. Fifth, we repurpose a mathematical model of glutathione metabolism to investigate how a single EBML/eSTR affects phenotype. Overall, our results leave little doubt that some polymorphic microsatellites contribute to different gene expression and phenotype in human populations.

## Results

### 313 EBML are associated with gene expression

We previously investigated 316,147 microsatellites across five super-populations (ethnicities) in the 1000 Genomes Project: African, East Asian, South Asian, European, and American [21]. We identified 3,984 ethnically biased microsatellite loci (EBML); for each EBML, at least one ethnicity has genotype frequencies statistically different from the remaining four. Since then, an independent study [10] of 175,226 short tandem repeats (STRs) identified 28,375 associated with nearby gene expression (eSTRs). Overlapping regions considered in both studies number 24,304 (Fig 1a).

We construct a 2x2 contingency table based on classifications (eSTR and/or EBML) of the 24,304 shared regions (Fig 1b). The overlap is statistically significant (p = 6.3e-13; $\chi^2$ test). Overall, we identify 313 overlapping EBML/eSTRs including 36 (out of 64) identified in our

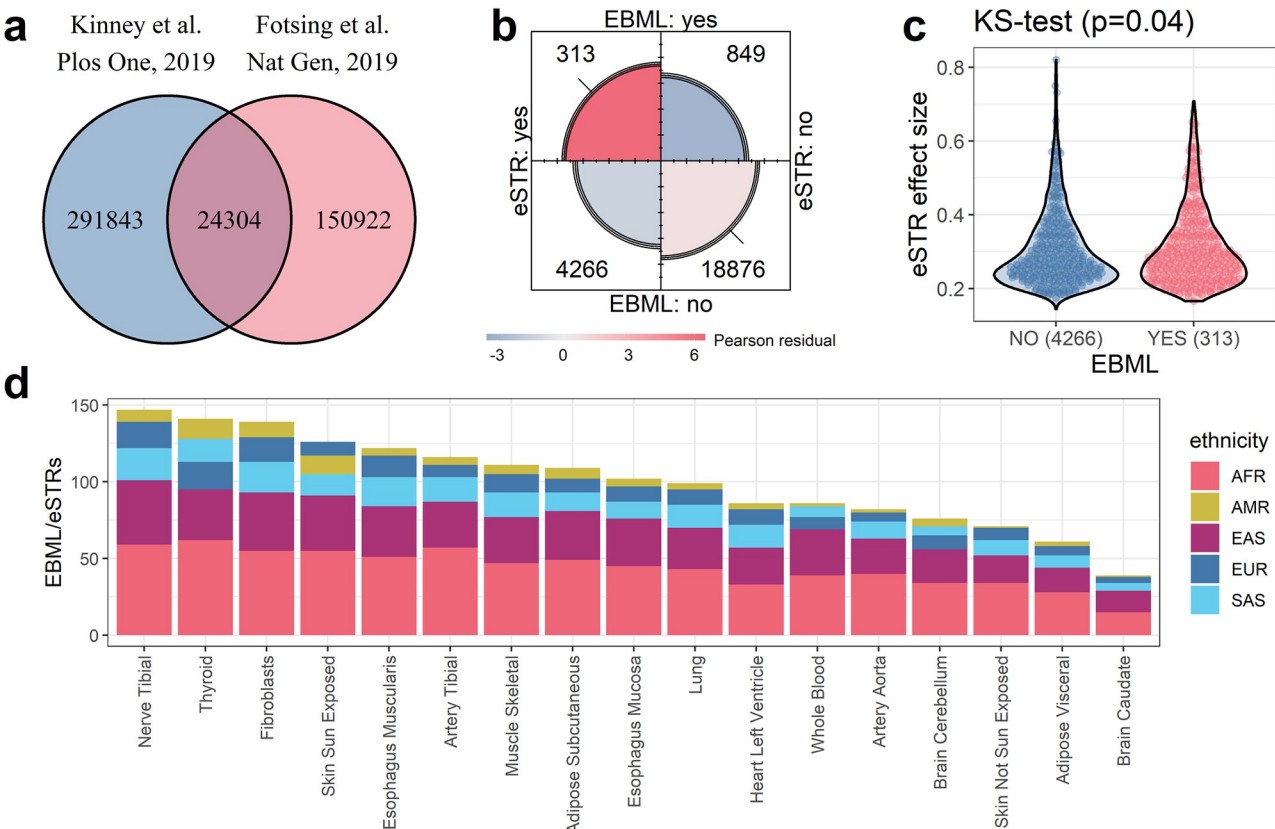

**Fig 1. Characterization of overlapping EBML/eSTRs. a**, We consider the set intersection of microsatellites from studies of ethnically biased microsatellites (EBML) and expression STRs (eSTRs): 24,304 microsatellites total. **b**, Fourfold plot reveals statistically significant overlap between EBML and eSTRs. Area of each quarter circle is proportional to the cell frequency (following marginal standardization). Color of each quarter circle corresponds with its Pearson residual: red indicates the cell entry exceeds the expected value; blue indicates less than then expected value. Confidence rings for the odds ratio allow a visual test of the null (no association); here, 99% confidence rings do not overlap indicating the null is rejected. **c**, Overlapping EBML/eSTRs tend to have larger effect sizes (effect on gene expression) than all other eSTRs. **d**, Characterization of the 313 overlapping microsatellites by ethnicity and tissue type. Note, that each microsatellite is biased in one or more ethnicities; and, each affects gene expression in one or more tissue types.

previous study [21]. The 313 regions provide preliminary support to our hypothesis that some microsatellites contribute to differential gene expression in human populations. Their preliminary support stems from a two-fold observation: (a) genotypes for each region differ by ethnicity and (b) genotypes for each region affect gene expression. The same cannot be said for the remaining 23,991 regions. In the results that follow we characterize the 313 candidate regions across ethnicity and tissue type; then, use RNA-seq data to pursue validation for a subset of 49 regions (out of the 313) associated with whole blood gene expression.

## Overlapping EBML/eSTRs characterized by ethnicity and tissue type

The 24,304 STRs considered in our analysis are associated with 17 different tissue types. This enables us to test EBML/eSTR overlap in each tissue and characterize the 313 candidate regions across ethnicity and tissue type. We begin by constructing similar 2x2 contingency tables for each tissue based on repeat classification (eSTR and/or EBML). The number of overlapping EBML/eSTRs varies for each tissue; nevertheless, each $\chi^2$ test of association reiterates significant mutual overlap (S1 Fig). The 313 candidate regions include abundant overlapping EBML/eSTRs associated with nerve tibial and thyroid genes (Fig 1d). Despite the abundance

of EBML in Africans and Asians (Fig 1d), samples from these populations are likely under-represented in the KGP data [22,23]. Thus, the true number of overlapping EBML/eSTRs for these populations may be much larger than what is shown in Fig 1d and S3 Data.

Next, we compare the previously measured effect sizes for sets of eSTRs based on their overlap with EBML (yes or no). Essentially, effect size (β) quantifies impact on gene expression (see methods). Interestingly, we find a difference in effect size for the two sets of eSTRs; the magnitude of effect size tends to be slightly larger for overlapping EBML/eSTRs (Fig 1c). The difference is statistically significant (p = 0.04; two sided Kolmogorov-Smirnov test).

We reiterate that the 313 overlapping EBML/eSTRs provide preliminary support to our hypothesis that some microsatellites contribute to differential gene expression in human populations. At the same time, we recognize the possibility that EBML are simply in linkage disequilibrium with targets of selection such as nearby SNPs. In fact, the 313 EBML/eSTRs have already been scrutinized to infer possible hitcher effects. Essentially, causality of eSTRs was estimated using CAVIAR: a type of statistical fine-mapping [24]. On average, 12.2% of eSTRs had higher causality scores than all nearby SNPs [10]. Our subsequent analysis seeks stronger support for our hypothesis with RNA-seq data. We reason that the 313 candidate regions can be validated by showing that associated genes are indeed differentially expressed in at least two ethnicities. We pursue validation with whole exome lymphoblastoid RNA-seq of 89 Africans and 373 Europeans (see methods). Validation is focused on the 54 genes associated with 49 overlapping whole blood EBML/eSTRs.

## EBML/eSTRs contribute to differential expression of 32 genes

We use RNA-seq to compare whole exome gene expression in cohorts of 89 Africans and 373 Europeans (see methods). We identify 14,124 (out of 60,667) differentially expressed genes (red and blue regions in Fig 2a and 2b) and cross reference with the 54 genes associated with whole-blood EBML/eSTRs. We find 32 (out of 54) genes associated with these EBML/eSTRs are indeed differentially expressed. We construct a 2x2 contingency and perform a test of the null: the number of differentially expressed genes associated with overlapping EBML/eSTRs (32 out of 54) does not exceed all other genes (14,124 out of 60,667). The null is rejected (p = 1.08e-9; $\chi^2$ test). We conclude that overlapping EBML/eSTRs are associated with changes in gene expression which in turn supports our overall hypothesis that microsatellites contribute to differential gene expression.

Next, we assess the direction of fold change for the 32 validated genes. First, all 14,124 differentially expressed genes are tabulated by fold change direction (up or down) and association with an EBML/eSTRs (yes or no). We identify 8,640 upregulated genes including 20 associated with an overlapping EBML/eSTR (Fig 2b). The 5,484 downregulated genes include 12 associated with an overlapping EBML/eSTR. The null was not rejected (p = 1; $\chi^2$ test) indicating that EBML/eSTRs are not disproportionately associated with upregulation or downregulation. The two most upregulated genes include MAEL and CYB5R2 with expression fold change approximately 1.75 for each. The two most downregulated genes are GPX7 and GSTM3; expression fold changes are .73 and .58, respectively. The latter two genes are involved in glutathione metabolism. In the results below we revisit these genes–GPX7 in particular–to explore potential effects on phenotype.

Superficially, these results suggest that EBML/eSTRs make a rather small contribution to overall differential gene expression; we verify just 32 affected genes out of 14,124. However, many EBML and eSTRs remain undiscovered which renders the true number of affected genes difficult to infer. A more reliable–and interesting–conclusion stems from the observation that EBML/eSTRs are not biased towards gene upregulation or downregulation. This is a bit

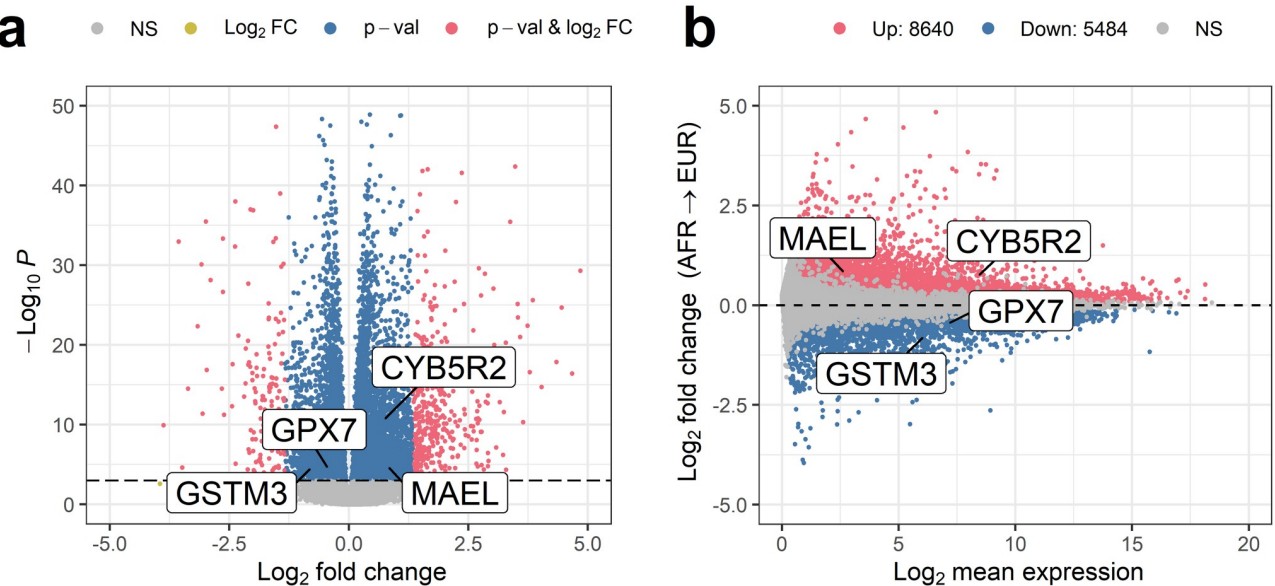

**Fig 2. Gene expression inferred from whole blood RNA-seq in cohorts of 89 Africans and 373 Europeans. a**, Volcano plot reveals 14,124 (out of 60,667) differentially expressed genes. Differential expression is validated for 32 (out of 54) genes associated with an EBML/eSTRs (p = 1.08e-9; $\chi^2$ test). Genes up regulated by EBML/eSTR microsatellites include MAEL and CYB5R2; genes down regulated by EBML/eSTR microsatellites include GPX7 and GSTM3. **b**, MA plot of gene expression fold change in 89 Africans and 373 Europeans. The 8,640 upregulated genes include 20 associated with an overlapping EBML/eSTR; the 5,484 downregulated genes include 12 associated with an overlapping EBML/eSTR. The null was not rejected (p = 1) indicating that EBML/eSTRs are not disproportionately associated with up regulation or down regulation.

surprising since microsatellites induce changes in gene expression through their unique mechanism of array length polymorphism. One might expect their effects to be distinguishable from other forms of gene regulation but apparently this is not the case. However, this does not rule out the possibility that effects of array length polymorphisms are distinguishable with respect to the magnitude of gene regulation. We explore this possibility with additional analysis.

## EBML/eSTRs have small effects on gene expression fold change

We compare the distribution of expression fold changes for differentially expressed genes based on their association with an overlapping eSTR/EBML (Fig 3a) and perform a test of the null: no difference in magnitude of fold changes. The null is rejected suggesting that expression fold change for differentially expressed genes affected by eSTRs/EBML tends to be less than all other differentially expressed genes (p = 0.0024; two sided Kolmogorov-Smirnov test). To be sure, we repeat the test for genes based on their association any eSTR. Once again, the null is rejected indicating that differentially expressed genes associated with eSTRs have relatively small changes in expression. Apparently, the effects of EBML/eSTRs are unbiased with respect to the direction of gene regulation (previous section) and biased with respect to the magnitude of gene regulation.

We recognize that expression fold change for genes associated with EBML/eSTRs does not necessarily reflect their biological impact or contribution to population level differences in gene expression. We partially address these limitations with principal component analysis (PCA). PCA analysis of differentially expressed genes verifies the presence of two population clusters corresponding to the cohort of 89 Africans and 373 Europeans: red and blue clusters in Fig 3c. Next, we compare the magnitude of PCA loadings for genes based on their association with an eSTR/EBML (Fig 3d and 3e). A test of the null reveals no difference in magnitude

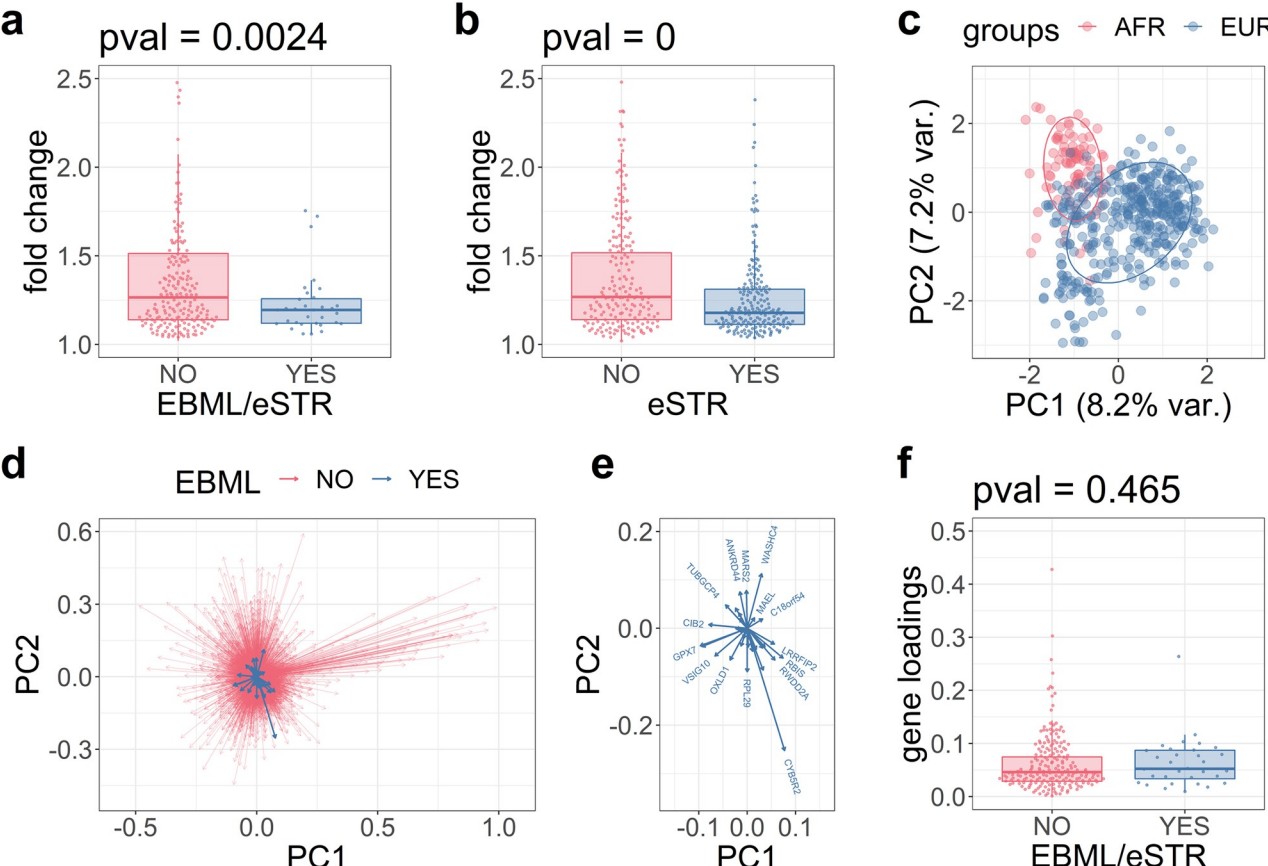

**Fig 3. The contribution of overlapping EBML/eSTRs to differential gene expression. a**, fold change for differentially expressed genes associated with EBML/eSTRs (blue) compared to all other genes (red). The difference is statistically significant (p = .0024; two sided Kolmogorov-Smirnov test). **b**, fold change for differentially expressed genes associated with eSTRs (blue) compared to all other genes (red). The difference is statistically significant (p = 0; two sided Kolmogorov-Smirnov test). **c**, PCA plot of differentially expressed genes reveals two population clusters: African samples (red) and European samples (blue). **d**, PCA loadings for differentially expressed genes including those associated with EBML/eSTRs (blue) and all other genes (red). **e**, inset of PCA loadings for genes associated with EBML/eSTRs. **f**, a comparison of PCA loadings finds no significant difference for genes association with EBML/eSTRs vs all other genes (p = .465; two sided Kolmogorov-Smirnov test).

of PCA loadings (p = .465; two sided Kolmogorov-Smirnov test). We conclude that genes associated with EBML/eSTRs are consistent with all other differentially expressed genes in terms of their contribution to PCA clusters (Fig 3f).

Although the eSTRs we consider are previously validated, our results do not imply that any changes in observed gene expression are truly caused by repeat array length polymorphisms. In any case, we reason that a better understanding of differential expression can be reached by performing regression analysis of the 32 validated genes associated with 32 EBML/eSTRs.

## 15 genes are associated with EBML/eSTR length polymorphisms

We perform regression analysis of repeat length vs gene expression for the 32 validated genes associated with 32 overlapping eSTR/EBML (see methods). The association is significant for 15 (out of 32) genes (Fig 4). The direction of array length change is split with respect to increasing and decreasing gene expression. In eight cases, array length insertions correlate with a proportional increase in gene expression (β>0); the remaining seven correlations are negative (Fig 4). We could not establish a correlation between repeat length and gene

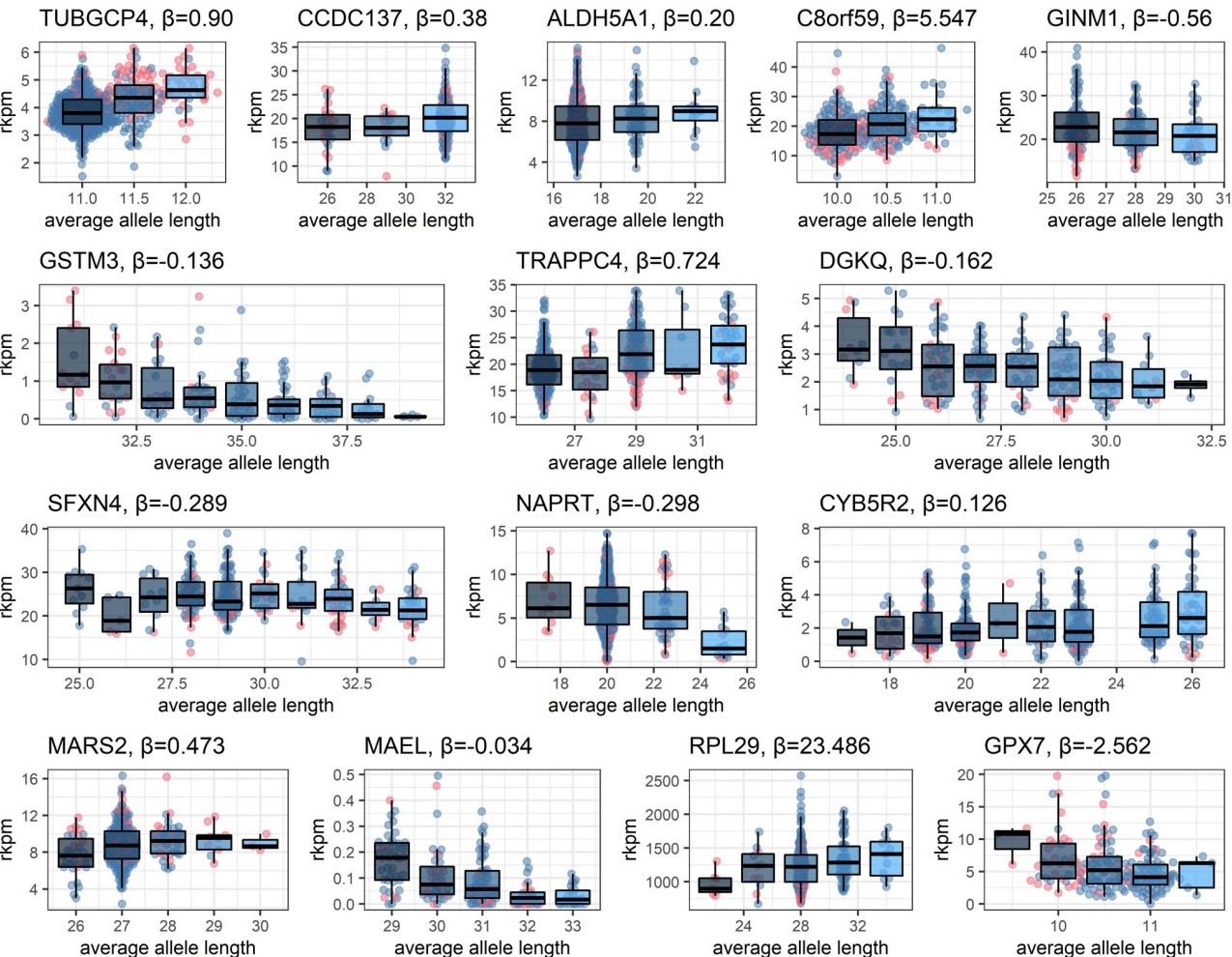

**Fig 4. Array length polymorphisms for 15 microsatellites correlate with gene expression.** Each gene is associated with an overlapping EBML/eSTR. In each case the null is rejected: no association between repeat length and gene expression. African samples (red) and European samples (blue) visually reiterate that array length polymorphisms are ethnically biased. Enrichment analysis reveals 1 significant KEGG pathway: glutathione metabolism. Genes involved in glutathione metabolism include GSTM3 and GPX7.

expression for 17 EBML/eSTRs (S2 Fig). However, it would be premature to say these "failed" associations are inconsistent with previous results. One possibility is that we are simply under-powered to reiterate these specific eSTR associations.

Interestingly, three of the positive gene associations were negative in a previous study of eSTRs [10]: ALDH5A1, RPL29, and C8orf59. These results seem to be mutually inconsistent; however, there is no known biology prohibiting the direction of eSTR associations from changing. Another explanation could stem from genetic differences in the groups of samples we and others use in regression analysis, i.e. Simpsons' paradox. Regardless, the 15 statistically significant EBML/eSTRs are consistent with four observations: (a) genotypes differ by ethnicity; (b) genotypes affect gene expression; (c) associated genes are differentially expressed; and (d) array length polymorphisms correlate with expression of associated genes. This leaves little doubt that these microsatellites contribute to differential gene expression in human populations.

Indirect links to phenotype are made by performing KEGG enrichment analysis on the set of 15 genes with verifiable links to overlapping EBML/eSTRs. We identify a single statistically

significant pathway: glutathione metabolism (p = .036). Statistical significance seems to be on tenuous footing with just two genes involved in this pathway: GSTM3 and GPX7. Perhaps confidence in the result is bolstered by recalling that GSTM3 and GPX7 were the most downregulated among 32 genes associated with overlapping EBML/eSTRs (Fig 2). Fortuitously, the effects of GPX7 have been the subject of intense mathematically modeling [25]. We use a well-established model of glutathione metabolism to complete the predictive links between repeat length, differential GPX7 expression, and phenotype [25].

### An eSTR/EBML associated with GPX7 affects oxidative stress

We use the Reed model of glutathione metabolism [25] to predict the effects of an eSTR (chr1:52525366) on GPX7 expression which in turn effects phenotype; in particular, response to oxidative stress. Briefly, the Reed model incorporates one carbon metabolism (methionine and folate cycles) as well as the transsulfuration pathway, the synthesis of glutathione, and its transport into the blood (Fig 5a). Activity of GPX is described by a modified Michaelis Menten term. There are no parameters representing expression of the GPX gene; nevertheless, we reason that the maximum reaction rate parameter ($GPX_{vm}$) can be used for this purpose.

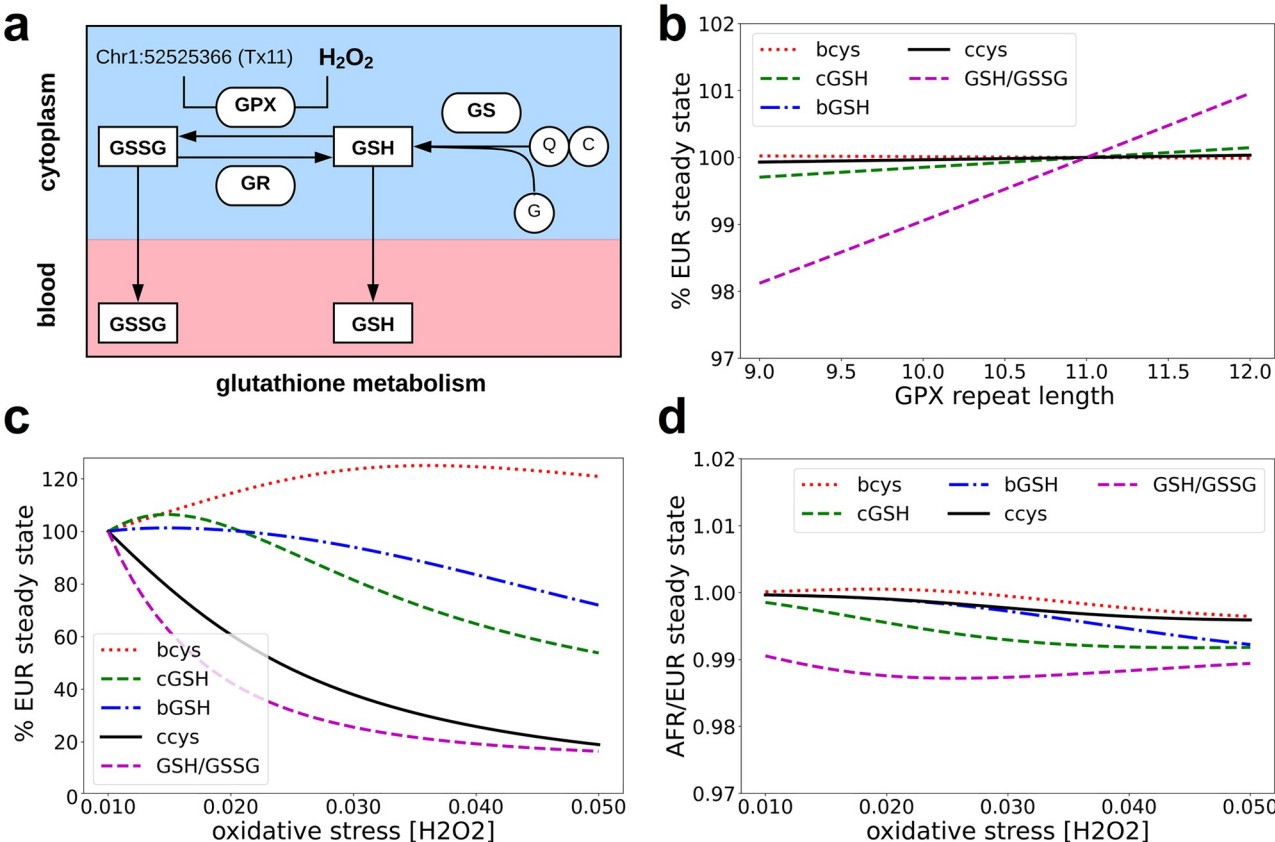

**Fig 5. A mathematical model of glutathione metabolism predicts differential response to oxidative stress in Africans and Europeans. a,** We modify the Reed model of glutathione metabolism to include effects of an eSTR (chr1:52525366) associated with GPX7 expression. Subsequent in silico experiments track the ratio [GSH]/[GSSG] indicative of the redox status of the cell. **b,** Predicted steady state concentrations of key metabolites are affected by eSTR array length. Array length deletions (shorter than 11bp) mimic the effects of oxidative stress. **c,** The European eSTR allele run to steady state under various concentrations of $H_2O_2$ faithfully reproduces the original Reed model. **d,** Steady state concentrations predicted for the African model (10bp allele) under various concentrations of $H_2O_2$ reveal differences in the oxidative stress response.

Formally, $GPX_{vm}$ represents the reaction rate at GPX saturation; therefore, increase or decrease in $GPX_{vm}$ must stem from a proportional increase or decrease in gene expression, respectively.

To study the effects of the specific eSTR (chr1:52525366) associated with GPX7 expression we promote $GPX_{vm}$ to a linear function of eSTR repeat length:

$$GPX_{vm} \rightarrow GPX_{vm}(1 + \beta_{eSTR}X/\lambda)$$

Here $X$ denotes eSTR array length deviation from the reference genome, $\beta_{eSTR}$ denotes its effect size, and $\lambda$ is the contribution of GPX7 to the GPX enzyme pool. The value of $\beta_{eSTR}$ (-0.30) is taken directly from experiment [10]; the value of $\lambda$ (.02) is derived from RNA-seq (see methods for details). We use the modified Reed model to predict effects of 9, 10, 11, and 12 base pair eSTR alleles, respectively.

We begin with the 11bp (reference) allele. Its deviation from the reference is trivially zero ($X = 0$); consequently, our modified Reed model reduces to the original Reed model. Nevertheless, we run the model to study state under varying concentrations of $H_2O_2$ which faithfully reproduces Reed's original 2008 results (Fig 5c). Except for blood cysteine, oxidative stress lowers the steady state concentrations of tracked metabolites (Fig 5c). We recall that the 11bp allele is common outside of Africa and refer to these as "European" results (Fig 5c).

The 10bp eSTR allele is common among Africans. Here, deviation from the reference is one ($X = -1$) and $GPX_{vm}$ proportionally increases. Note consistency with our own regression analysis (Fig 4; lower right). The "African" results are generated by running this model to its new steady state under normal (.01μm) $H_2O_2$ concentration. The African (10bp) allele lowers the steady state ratio of reduced/oxidized glutathione concentrations (Fig 5b). With respect to these metabolites, the African (10bp) allele appears to mimic the European (11bp) allele under oxidative stress (Fig 5b and 5c). In addition, the Reed model–modified by the African allele– produces a different response when the concentration of $H_2O_2$ is varied and run to steady state (Fig 5d). The ratio of reduced/oxidized glutathione concentration remains lower in the "African model" regardless of $H_2O_2$ concentration (Fig 5d).

We run two additional models based on 9bp and 12bp minor alleles. Our previous analysis finds the former allele in Africans and non-Africans; the latter allele is exclusive to non-Africans (see methods). The ratio of reduced/oxidized glutathione further decreases in response to the 9bp allele and increases in response to the 12bp allele (Fig 5b).

Our approach to modeling the effects of an eSTR (chr1:52525366) associated with GPX7 expression is admittedly imperfect. A myriad of additional details likely affects the oxidative stress response in Africans and Europeans. Qualitatively speaking, predictions of the modified Reed model appear to be consistent with established literature. For example, heightened oxidative stress has been reported for African Americans *in vitro* and *in vivo* [26]; and, decreased GPX activity has been directly measured in various South African populations [27]. We conclude by reiterating the predictive link between repeat length, differential GPX7 expression, and phenotype. Repeat length (eSTR at chr1:52525366) is inversely associated with GPX7 expression; consequently, GPX7 expression and the ratio [GSH]/[GSSG] tends to be lower in Africans compared to Europeans.

## Discussion

Microsatellite research has broadly pursued two lines of inquiry. Older studies used microsatellites to probe human population structure [12–15]; a pair of recent studies addressed how microsatellites affect gene expression [9,10]. These two lines of inquiry may share a link; in particular, we recently discovered significant overlap between ethnically biased microsatellite loci (EBML) and expression short tandem repeats (eSTRs). However, we fell short of validating

or quantifying the contribution of the overlapping regions to differential gene expression. Here we narrow the gap using a combination of known EBML, newly discovered eSTRs, and RNA-seq. Our main results are threefold. First, we identify 313 overlapping EBML and eSTRs. Second, whole blood RNA-seq confirms that 32 (out of 54) associated genes are differentially expressed in Africans and Europeans. Third, a repurposed mathematical model of glutathione metabolism predicts that array length polymorphisms in one of these affects cellular response to oxidative stress. Overall, our results support the hypothesis that some microsatellites contribute to differential gene expression and phenotype in human populations.

Our findings greatly expand our previous results [21]. Indeed, a set of 3,984 EBML and 28,375 newly discovered eSTRs show significant overlap; the 313 overlapping regions expand the set of 64 identified in our previous study. The discovery of additional polymorphic regions (EBML) that affect gene expression (eSTRs) supports our hypothesis and motivates the remainder of our study. We further expand our previous results by characterizing the 313 overlapping regions across 17 tissue types. Overlapping EBML/eSTRs are numerous in nerve tibial, thyroid, and fibroblasts. This is intriguing given the established role of microsatellites in various neurodegenerative diseases; however, without nerve tibial gene expression data we were unable to rigorously verify that affected genes are differentially expressed. Fortunately, lymphoblastoid gene expression data was available for cohort of 89 Africans and 373 Europeans [28]. Thus, our subsequent results focused on quantifying the contribution of microsatellites to whole blood differential gene expression.

Our analysis of differential gene expression in whole blood supports the catchphrase that microsatellites are tuning knobs of gene expression [18,19]. We quantify this catchphrase by comparing expression fold change for differentially expressed genes in Africans and Europeans; the magnitude of change is significantly smaller for genes affected by EBMLs and eSTRs. Based on this finding, we propose that microsatellites polymorphisms have relatively small effects on gene expression. This is speculative; however, we draw a modicum of support from linear regression analysis of microsatellite allele length in relation to reads per kilobase of transcript (RPKM). Our results–and previous results [10]–find most eSTRs have regression coefficients less than one. In other words, the fine-tuning effects of small changes in eSTRs are almost undetectable using mapped reads. For this reason, we anticipate that higher coverage DNA and RNA sequencing will be needed to better understand the human repeatome and its effects on phenotype.

Effects of microsatellites on phenotype are typically inferred from genome wide association studies. Perhaps the largest study to date discovered that microsatellites contribute to the missing heritability of autism spectrum disorders [29]. The phenotypic link was supported in part by demonstrating enrichment of de novo microsatellite mutations in fetal brain promoters including a CAG repeat embedded in the 5'UTR of HDAC2. Incidentally, CAG repeats were recently associated with three genes potentially involved in medulloblastoma: RAI1, BCL6B, and TNS1 [30].

We develop a simple way to predict how microsatellites associated with phenotype take effect and demonstrate our approach by repurposing an existing mathematical model of glutathione metabolism [25]. In the original 2008 model, oxidation of glutathione via glutathione peroxidase was assumed to follow Michaelis-Menten kinetics. However, it is now known that an eSTR (chr1:52525366) affects expression of the glutathione peroxidase gene. We predict phenotypic effects of this eSTR by promoting the Michaelis-Menten $V_{max}$ constant to a function of repeat length with parameters taken directly from experiment. Our results suggest that the African and European alleles contribute to predictable differences in response to oxidative stress. More generally, this may be an easy way of predicting the phenotypic effects of other

eSTRs. After all, there are over 28,000 eSTRs and nearly 1,000 biological systems models for human available in databases such as JWS online and EBI BioModels.

Differences in African and European response to oxidative stress is backed by several studies. Our specific prediction–lower levels of GSH in African Americans–is supported by at least 6 studies [31–36]. Most of these focus on racial disparities in diabetes and cardiovascular disease which further bolsters significance of our predictions. African American diabetic patients have lower levels of GSH and lower circulating hydrogen sulfide then their European counterparts [31,32]. Differences in GPX activity in healthy cohorts have been measured in various South African populations [27]. More generally, reduced glutathione has been associated with various cancers, neurodegenerative diseases, Parkinson's disease, and cystic fibrosis [37]. In fact, there are timely links to COVID-19 severity which disproportionately affects African Americans; in particular, reduced glutathione is thought to lower viral load and viral infection [38–40]. Needless to say, thorough investigation of these lines of evidence is beyond the scope of this work.

The strength of our study may also be its weakness. Specifically, we use multiple datasets to show that microsatellites contribute to differential gene expression and phenotype in human populations; however, this leads to attrition in the number of microsatellites we investigate. We begin with two large genome wide surveys of 316,147 and 175,226 microsatellites [10,21], respectively. Only 24,304 overlapping microsatellites are considered in our subsequent analysis. We subsequently focus on 313 overlapping EBML and eSTRs; among these, only 49 can be investigated using whole blood RNA-seq. We pare down to a single microsatellite prior to investigating its effects on glutathione metabolism. Despite this attrition, our results contribute to what is known about racial/ethnic differences in oxidative stress. We conclude with a testable prediction: the 10bp African allele for an eSTR (chr1:52525366) increases expression of glutathione peroxidase which lowers reduced blood glutathione. This underscores a need to better understand the function of microsatellites and the human repeatome at large.

Several additional caveats stem from the datasets we consider. EBML were previously identified from the KGP samples which likely under-represent African and Asian diversity [22,23]. In addition, microsatellite allele calls remain challenging; particularly from NGS low coverage data. Thus, many EBML remain undiscovered and some of the known EBML are likely false positives. The differentially expressed genes–identified using RNA-seq data–undoubtedly also contain false positives due to random sampling inherent in the assay. Thus, more work needs to be done to validate these results experimentally.

## Methods

### Overall approach

Broadly speaking, we combine three existing datasets (red in Fig 6) to show that some microsatellites contribute to differential gene expression and phenotype in human populations:

(a). list of EBML and genotypes in 2529 samples from the 1000 genomes project [21]

(b). list of eSTRs including their effect sizes and associated genes [10]

(c). lymphoblastoid RNA-seq of 462 samples from the 1000 genomes project [28]

We begin by identifying and characterizing the overlap of datasets (a) and (b). The third dataset (c) is used as validation. Subsequent pathway enrichment analysis motivates predictive modeling of glutathione metabolism. The remainder of our methods (and results) unfold in

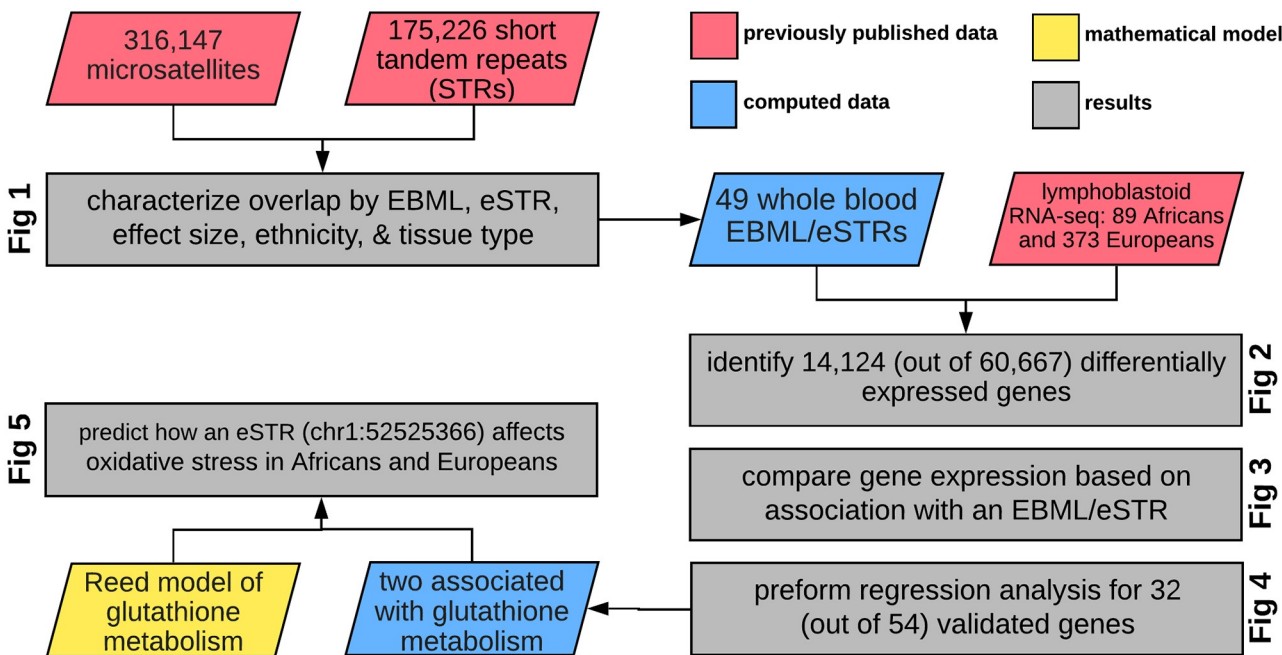

**Fig 6. Summary of our overall approach.** We characterize 24,304 overlapping regions drawn from two previous studies of short tandem repeats. We validate 32 (out of 54) genes associated with whole blood EBML/eSTRs using RNA-seq: 89 Africans and 373 Europeans. We subsequently quantify their effects on gene expression and perform KEGG pathway enrichment analysis. Two of the validated EBML/eSTRs affect glutathione metabolism. We modify the Reed model to predict how an eSTR associated with GPX7 expression affects response to oxidative stress in Africans and Europeans, respectively.

this general order with numerous additional details. Schematic of our overall approach is outlined in Fig 6.

## Microsatellite lists and test of EBML/eSTR association

We use precomputed lists of EBML and eSTRs available at PubMed identifier (PMID) 31830051 and 31676866, respectively [10,21]. Analysis of 3,984 EBML can be found in supplementary table 1 and S2 Data of the former publication [21]; analysis of eSTRs can be found in S1 and S2 Data of the latter publication [10]. Additional information regarding EBML and eSTRs are available online at http://www.cagmdb.org and http://www.webstr.ucsd.edu, respectively. Repeat regions considered in both publications– 24,304 total–were identified and characterized by cross-referencing the genome coordinates provided by these datasets. Our results are summarized in Fig 1; raw data is provided in S1 Data.

We constructed a 2x2 contingency table based on classifications of the 24,304 shared repeats (Fig 1b): EBML; not EBML; affects gene expression (eSTR); does not affect gene expression (other STR). We performed a $\chi^2$ test of the null: no association between EBML and eSTRs. The null was rejected (p = 6.3e-13), see results.

## Microsatellite genotyping

We use the microsatellite genotypes provided in S1 Data of our previous publication [21]; novel genotyping of microsatellites was not performed in this study. Nevertheless, we reiterate the procedure used previously. Briefly, we used RepeatSeq to determine the genotype of microsatellites in next generation sequencing reads for 2,529 samples in the 1,000 genomes project [41]. RepeatSeq operates on three input files: a reference genome, a file containing reads

aligned to the human reference genome (.bam file), and a list of query microsatellites. Repeat-Seq has been used in previous studies and is freely available: https://github.com/adaptivegenome/repeatseq. Additional details of microsatellites genotyping are provided in our previous publication [21].

We acknowledge that Repeatseq is no longer the industry standard for microsatellite genotyping in general. Several recent programs report similar or better accuracy: GangSTR [42], HipSTR [42], lobSTR [43], STRetch [44], TREDPARSE [45], and Dante [46]. A key advantage of GangSTR is the ability to genotype microsatellites that exceed the read length [42]. HipSTR is the industry standard for detecting shorter polymorphisms [42]. Our use of RepeatSeq was justified in our previous publication. In particular, RepeatSeq was specifically designed and validated using data from the 1000 Genomes Project [41]. In addition, RepeatSeq performs local realignment and multiple sequence alignment of microsatellite arrays. To ensure reproducibility of our results, all aligned reads used to infer genotypes have been published and archived online: http://www.cagmdb.org. See our previous work for a further discussion of the challenges inherent in microsatellite allele calling [21].

## Samples

Samples used to identify EBML and eSTRs can be found in previous publications. We use a third set of 462 samples to identify differentially expressed genes in a cohort of 89 Africans and 373 Europeans [28]. We briefly reiterate the source and availability of each sample set.

(a). *Discovery of EBML*. Samples used for discovery of EBML are documented in a previous publication [21]. Briefly, these samples come from phase 3 of the 1000 Genomes Project: ftp://ftp.1000genomes.ebi.ac.uk/vol1/ftp/phase3/. In total, 2,529 samples were included for analysis: 667 African (AFR), 502 European (EUR), 352 American (AMR), 514 East Asian (EAS), 494 South Asian (SAS).

(b). *Discovery of eSTRs*. Samples used for discovery of eSTRs are documented in a previous publication [10]. Briefly, authors of that study obtained next-generation sequencing data from GTEx through dbGaP (phs000424.v7.p2). They obtained gene-level RPKM values for 17 tissues from dbGaP (phs000424.v7.p2). Additional details can be found elsewhere.

(c). *Differentially expressed genes*. Differentially expressed genes were identified using publicly available RNA sequencing (RNA-Seq) data for a cohort of 462 lymphoblastoid cell lines from the 1000 genomes project [28]. These samples include 89 Africans and 373 Europeans. All samples are available online through the European Bioinformatics Institute website: https://www.ebi.ac.uk/arrayexpress/experiments/E-GEUV-1/samples/. Note that next generation (DNA) sequencing of these samples is available in phase 3 of the 1000 Genomes Project (see above).

## RNA sequencing (RNA-seq) data analysis

RNA-seq data analysis was performed in four steps: (a) quality control and preprocessing; (b) Alignment to the human reference genome; (c) read counting; and (d) differentially expression analysis.

(a). *Quality control and preprocessing*. Read quality reports were prepared with FastQC [47]. We ignored sequence duplication warnings which are expected for highly expressed transcripts. We also ignored warnings triggered by a small number of mildly affected tiles, i.e. per tile sequence quality warnings. We ignored K-mer content failures arising from

random priming. Effects of random priming can be addressed with read count reweighting schemes but are only recommended for studies of alternative splicing and de novo inference of gene structure [48]; we only consider gene level counts.

(b). *Alignment to the human reference genome*. RNA sequencing reads were aligned to the human reference genome using STAR: Spliced Transcripts Alignment to a Reference [49]. STAR is well suited for mapping reads containing indels. Mapping was conducted with the GRCh38 human reference sequence and GENCODE annotation release 33 (gencode.v33.annotation.gtf).

(c). *Read counting*. Per sample reads mapping to genes were counted with HTseq [50]. Briefly, exon counts (—type = exon) were aggregated by gene (—idattr = gene_id) assuming no strand specific protocol (—stranded = no). Counts for each sample were subsequently normalized using reads per kilobase of transcript per million mapped reads (FPKM): we use the countToFPKM package available in R.

(d). *Differential expression analysis*. Differentially expressed genes in the 89 Africans and 373 Europeans were identified using DESeq2 [51]. Briefly, we construct a matrix from the per sample HTseq gene-level counts: one sample per column and one gene per row. Using the count matrix, the DEseq2 pipeline was used to automatically estimate size factors, estimate dispersion for each gene, and fit a generalized linear model.

## Identification of differentially expressed genes, PCA analysis, and pathway enrichment

Differential gene expression analysis was performed using DESeq2 (see previous section). We subsequently identified 14,124 differentially expressed genes: Benjamini-Hochberg adjusted p-value less than .05.

In subsequent steps (see results), we perform principal component analysis (PCA) of differentially expressed genes in Africans and Europeans. We first applied a variance stabilizing transform (getVarianceStabalizedData function) which normalizes counts by division of size factors. PCA analysis was performed in R (prcomp). Details can be found elsewhere; briefly, the calculation is done using singular value decomposition of the data matrix.

Various gene sets throughout this work were tested for functional enrichments using the Kyoto Encyclopedia of Genes and Genomes (KEGG). We used the clusterProfiler package in R [52]–enrichKEGG function–to test each gene set and KEGG pathway with respect to the null hypothesis: pathway genes are not over-represented in the set of query genes. We used Benjamini-Hochberg adjusted p-vales. The null was rejected ($p$-value $< .05$) for glutathione metabolism using a query set of 15 genes (see results).

## Mathematical modeling of oxidative stress

We used a previously published mathematical model of glutathione metabolism [25] to predict the phenotypic effects of an eSTR (chr1:52525366) associated with expression of the glutathione peroxidase gene (GPX7). Parameters of the mathematical model–henceforth called the Reed model–can be found elsewhere [25]. Here we summarize the Reed model, detail a key modification to the GPX saturation curve, and describe our in-silico experimentation of oxidative stress.

(a). *Reed model of glutathione metabolism*. The self-described Reed model of glutathione metabolism is based on known properties of the enzymes and the regulation of those

enzymes by oxidative stress [25]. The model incorporates one carbon metabolism (methionine and folate cycles) as well as the transsulfuration pathway, the synthesis of glutathione, and its transport into the blood [25]. Rates of change for each substrate are described by 34 differential equations, 62 rection velocities, and 169 parameters. Despite the complexity, most of the parameters are taken directly from literature and most of the reaction velocities assume a standard Michaelis-Menten form for one or two substrates. A few reaction velocities in the original model were modified by an $H_2O_2$ inhibition or activation term. For example, the following reaction velocity contributes to the rates of change in reduced cytosolic glutathione (cGSH) and oxidized cytosolic glutathione (cGSSG):

$$\frac{GPX_{vm}[H_2O_2][cGSH]^2}{([H_2O_2] + 9GPX_{km}^{H_2O_2})(GPX_{km}^{gsh} + [cGSH])^2}$$

Here, concentrations of $H_2O_2$ and cGSH appear in square brackets; all other terms are model parameters. A full description of the Reed model is provided in the supplementary material of its publication [25]. The Reed model is also available online: https://jjj.bio.vu.nl/models/reed/.

(b). *Modification to the GPX saturation curve.* We modify the Reed model to study the effects of an eSTR (chr1:52525366) associated with expression of the glutathione peroxidase gene (GPX7). However, the Reed model does not contain a term describing expression of GPX7. Instead, we reason that the parameter $GPX_{vm}$ can be used a surrogate for GPX7 expression. Typically, $GPX_{vm}$ represents the maximum reaction rate occurring at GPX saturation. Thus, decreasing $GPX_{vm}$ indirectly represents decreasing GPX expression; likewise, increasing $GPX_{vm}$ indirectly represents increasing GPX expression. We assume changes in GPX expression are proportional to the $GPX_{vm}$ parameter. Then we promote the $GPX_{vm}$ parameter to a linear function of eSTR repeat length:

$$GPX_{vm} \rightarrow GPX_{vm}(1 + \beta_{eSTR}X/\lambda)$$

Here $X$ denotes eSTR array length deviation from the reference genome and $\beta_{eSTR}$ denotes its effect size. The value of $\beta_{eSTR}$ (-0.30) is taken directly from experiment. The value of $\lambda$ (.02) is derived from RNA-seq data (see below); it represents the contribution of GPX7 relative to all eight glutathione peroxidase genes (GPX 1–8). These modifications describe the effects of eSTR array length polymorphisms on GPX7 expression and in turn on the $GPX_{vm}$ parameter. Notice that array length insertions ($X>0$) decrease the maximum rate ($GPX_{vm}$) at which cytosolic glutathione is oxidized; on the contrary, array length deletions ($X<0$) increase the maximum rate ($GPX_{vm}$) at which cytosolic glutathione is oxidized. In the absence of array length polymorphism ($X = 0$) the $GPX_{vm}$ parameter is left unchanged and our modified Reed model reduces to the original Reed model. Additional details are provided in results. The original Reed model, the modified Reed model, and all in-silico experiments can be reproduced using a dockerized jupyter notebook provided in our supplementary material: see S1 Code.

(c). *In-silico experimentation of oxidative stress.* Our previous study of EBML identified 4 alleles for the eSTR (chr1:52525366) associated GPX7 [21]: 9, 10, 11, and 12 base pair alleles. Alleles for each sample in the 1000 genomes project are available online (www.cagmdb.org/view_table.php?id=226795). We used the modified Reed model of

glutathione metabolism (see above) to perform four series of in silico experiments: one for each allele. Each series was a two-step process. First, we determined the allele deviation from the reference genome ($X$), i.e., deviation for the 10bp allele is -1. Then, we ran the modified Reed model to steady state at a range of $H_2O_2$ concentrations (.01μm to .05μm).

Effects of the four GPX7 alleles were inferred by comparing steady state concentrations of various metabolites. Our results pay particular attention to the ratio [GSH]/[GSSG] which changes as reduced blood glutathione (GSH) sequesters the reactive oxygen species ($H_2O_2$). In general, this ratio is indicative of the redox status of the cell, has been shown to regulate redox sensitive enzymes, and plays a role in disease [53,54]. Our results also pay particular attention to the 10 and 11 base pair eSTR alleles. The former allele is common among African populations; the latter allele is common among European populations [21]. The *in-silico* experiments based on these alleles are assumed to be representative of the oxidative stress response in Africans and Europeans, respectively (see results).

### Derivation of λ

The GPX7 gene encodes one member in a family of eight isoenzymes (GPX 1–8). Therefore, effects of eSTR repeat length polymorphisms on GPX7 expression only affect a fraction of the entire GPX enzyme pool. We use an adjustable parameter (λ) to represent the fraction specifically derived from GPX7 expression. We estimate λ by comparing expression of all GPX genes in the RNA-seq cohort of 89 Africans and 373 Europeans (Fig 7). GPX1 and GPX4 are most abundant with average expression (rpkm) values of 4,290 and 4,096, respectively. Average expression (rpkm) for GPX7 is considerably lower: 159 (Fig 7). We conclude that GPX7 expression accounts for approximately 2% (λ = .02) of the entire GPX enzyme pool.

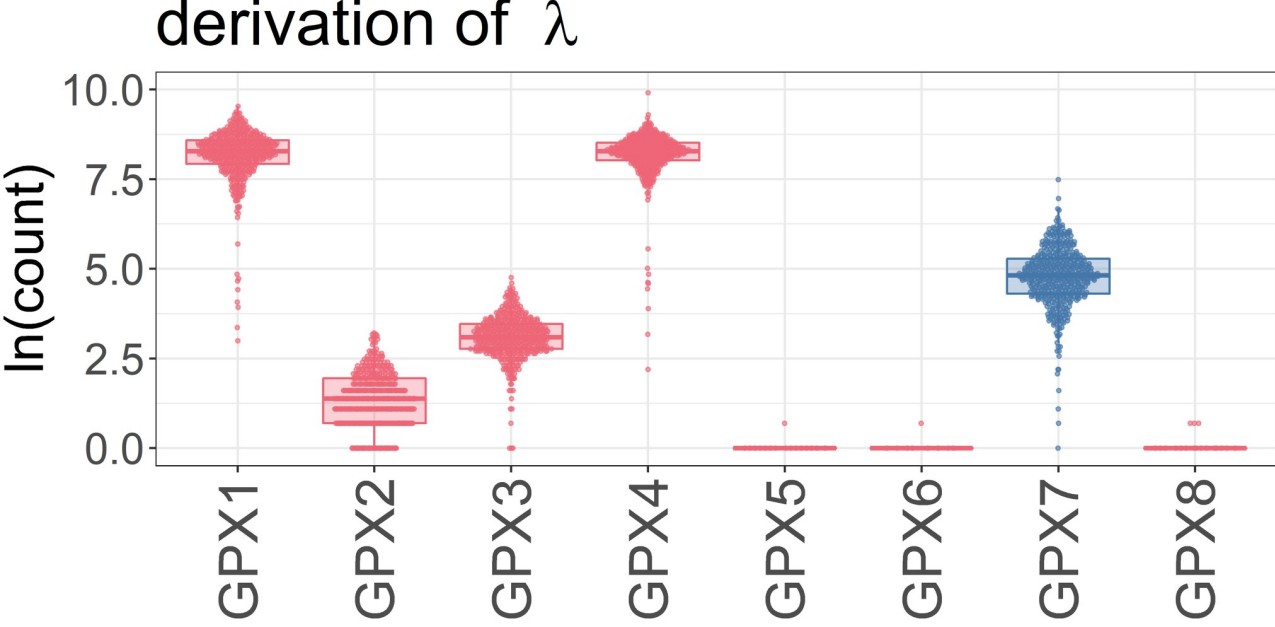

**Fig 7. GPX7 gene encodes one member in a family of eight isoenzymes (GPX 1–8).** We estimate the contribution of GPX7 to the entire GPX enzyme pool using RNA-seq. GPX7 expression accounts for approximately 2% (λ = .02) of the entire GPX enzyme pool.

## Regression analysis for repeat length vs gene expression

We checked for association between genotype and gene expression for 49 whole blood eSTRs associated with 54 genes in 462 samples: 89 Africans and 373 Europeans. Each microsatellite genotype (see above) was first recast as the array length average of both alleles. For example, a sample with 10 and 11 base pair alleles for the eSTR at chr1:52525366 received a 10.5 base pair genotype. Genotypes were then regressed with normalized gene expression (rpkm):

$$Y^{'} = \beta_{eSTR}X + \epsilon$$

The same approach has been used previously: Y' denotes normalized rpkm; X denotes microsatellite genotypes; β denotes effect size and ε is the error term. We used R (cor.test function) to perform a correlation test of the null: slope (β) equal to zero. The null was rejected for 15 genes (see results).

## Statistical tests of gene loadings and gene expression fold changes

The contribution of eSTRs to differential gene expression was quantified in three steps. First, differentially expressed genes (detailed above) were tabulated by their association with an eSTR (yes or no). Second, we compared the distribution of unsigned expression fold changes for the two sets of genes. Third, we performed a Kolmogorov-Smirnov test of the null: no difference in fold change. The null was rejected indicating that genes associated with eSTRs have relatively small changes in expression compared to all other differentially expressed genes (see Fig 3b).

We performed a similar test designed to take into consideration EBML. First, differentially expressed genes were tabulated by their association with an overlapping EBML/eSTR (yes or no). The remaining steps were left unchanged and the null was rejected (see Fig 3a).

We performed a third test designed to investigate whether overlapping EBML/eSTRs contribute to the distinct population clusters seen in PCA plots of 89 Africans and 373 Europeans. First, we performed PCA analysis of differentially expressed genes (details above). Next, we tabulated genes by their association with an overlapping EBML/eSTR (yes or no) and compared their PCA loadings; we used the magnitude of PCA loadings in the first and second principal components (Fig 3f). Finally, we performed a Kolmogorov-Smirnov test of the null: no difference in PCA loadings for genes associated with EBML/eSTRs compared to all other differentially expressed genes. The null was not rejected (Fig 3f).

The absence of a suitable null hypothesis prevented us from performing statistical tests for genes lacking differential expression. For example, genes lacking differential expression are easily tabulated by their association with an eSTR (yes or no); however, a rigorous null hypothesis is not available for testing whether these gene sets have differences in expression fold change.

## Novelty compared to our previous publication

Our previous publication (10.1371/journal.pone.0225216) in this field had four main results. First, we identified 3,984 ethnically biased microsatellite loci (EBML). Second, we showed that a statistically significant number of EBML coincide with known eSTRs; i.e. they affect gene expression. Third, we demonstrated association between EBML, core matrisome genes, and matrisome associated genes. Fourth, we found a significant number of EBML in putative selective sweep regions. Overall, these findings were observational in nature. This work pursues questions related to how EBML affect gene expression, i.e. questions that are functional in nature. How many EBML affect gene expression? What is the impact of EBML on gene

expression relative to other differentially expressed genes? Can we infer how EBML affect phenotype using mathematical models?

## Code availability

We used freely available R packages to perform our analysis. In addition to those already mentioned, we use fourfold plots from the visualizing categorical data (vcd) package. Volcano plots of gene expression were prepared with the EnhancedVolcano package. The gene level MA-plot was prepared with ggpubr (ggmaplot function). Remaining plots were prepared with ggplot2. The modified Reed model is available as a dockerized jupyter notebook (S1 Code). Results can be rerun in a dockerized jupyter notebook available on github: https://github.com/nkinney06/dockerizedEBML.

## Supporting information

**S1 Fig. Test of EBML/eSTR association in 17 tissue types.**
(TIF)

**S2 Fig. Regression analysis of 17 non-significant eSTRs.**
(TIF)

**S1 Data. Summary of 24,304 microsatellites considered in this study.**
(7Z)

**S2 Data. Effect sizes for EBML and eSTRs by tissue type.**
(TXT)

**S3 Data. 313 overlapping EBML/eSTRs characterized by ethnicity and tissue type.**
(TXT)

**S4 Data. RPKM vs repeat length for 54 genes in 89 Africans and 373 Europeans.**
(TXT)

**S5 Data. HTseq counts of whole exome gene expression in 89 Africans and 373 Europeans.**
(7Z)

**S1 Code. Dockerized jupyter notebook to reproduce all results including the modified Reed model of glutathione metabolism.**
(ZIP)

## Acknowledgments

NAK, LK, RA, PM, JC, and HRG contributed to the conceptualization of this project, experimental design, and data analysis. NAK, LK, RA, and PM contributed the writing of this manuscript. NAK and LK were responsible for software writing and data analysis. NAK, LK, HB, JC, EL, MH, KH, IS, YS, RA, PM, and HRG were responsible for manuscript preparation. All authors read and approved the final manuscript.

## Author Contributions

**Conceptualization:** Nick Kinney, Lin Kang, Javan K. Carter, Ramu Anandakrishnan, Pawel Michalak, Harold Garner.

**Data curation:** Nick Kinney, Lin Kang, Harpal Bains, Elizabeth Lawson, Mesam Husain, Kumayl Husain, Inderjit Sandhu, Yongdeok Shin, Ramu Anandakrishnan, Pawel Michalak, Harold Garner.

**Formal analysis:** Nick Kinney, Lin Kang, Harpal Bains, Elizabeth Lawson, Mesam Husain, Kumayl Husain, Inderjit Sandhu, Yongdeok Shin, Ramu Anandakrishnan, Pawel Michalak.

**Funding acquisition:** Nick Kinney, Harold Garner.

**Investigation:** Nick Kinney, Lin Kang, Harpal Bains, Elizabeth Lawson, Mesam Husain, Kumayl Husain, Inderjit Sandhu, Yongdeok Shin, Ramu Anandakrishnan, Pawel Michalak, Harold Garner.

**Methodology:** Nick Kinney, Lin Kang, Javan K. Carter, Ramu Anandakrishnan, Pawel Michalak, Harold Garner.

**Project administration:** Nick Kinney, Harold Garner.

**Resources:** Nick Kinney, Pawel Michalak, Harold Garner.

**Software:** Nick Kinney, Lin Kang, Harold Garner.

**Supervision:** Nick Kinney, Harold Garner.

**Validation:** Nick Kinney, Elizabeth Lawson, Javan K. Carter.

**Visualization:** Nick Kinney.

**Writing – original draft:** Nick Kinney, Ramu Anandakrishnan, Pawel Michalak, Harold Garner.

**Writing – review & editing:** Nick Kinney, Lin Kang, Harpal Bains, Elizabeth Lawson, Mesam Husain, Kumayl Husain, Inderjit Sandhu, Yongdeok Shin, Javan K. Carter, Ramu Anandakrishnan, Pawel Michalak, Harold Garner.

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
