## [Decision Letter · Decision Letter 0]

16 Feb 2021

PONE-D-21-01478

Ethnically biased microsatellites contribute to differential gene expression and glutathione metabolism in human populations

PLOS ONE

Dear Dr. Kinney,

Thank you for submitting your manuscript to PLOS ONE. After careful consideration, we feel that it has merit but does not fully meet PLOS ONE’s publication criteria as it currently stands. Therefore, we invite you to submit a revised version of the manuscript that addresses the points raised during the review process.

The reviewers are experts in the field and were overall positive about this manuscript. However, a few significant concerns were raised regarding the study design and its limitations. All comments below (from Editor and Reviewers) must be carefully addressed and incorporated in the revised manuscript accordingly.

We look forward to receiving your revised manuscript.

Kind regards,

Danillo G Augusto

Academic Editor

PLOS ONE

3. Please upload a copy of Supporting Information S1, S2, S3, S4, S5 Data and S1 Code  which you refer to in your text on page 30

.

**Additional Editor Comments:**

Please clarify the similarities and differences of this current submission compared to the previously published manuscript by the same authors: 10.1371/journal.pone.0225216. Please highlight the differences in the aims and also the novelty this current submission brings compared to the previous manuscript. If necessary, please amend the text to clarify this point.

**Reviewers' comments:**

**Comments to the Author**

1. Is the manuscript technically sound, and do the data support the conclusions?

Reviewer #1: Partly

Reviewer #2: Yes

2. Has the statistical analysis been performed appropriately and rigorously? 

Reviewer #1: Yes

Reviewer #2: Yes

3. Have the authors made all data underlying the findings in their manuscript fully available?

Reviewer #1: Yes

Reviewer #2: Yes

4. Is the manuscript presented in an intelligible fashion and written in standard English?

Reviewer #1: Yes

Reviewer #2: Yes

5. Review Comments to the Author

Reviewer #1: Kinney et al. sought to identify the contribution of EBML and eSTRS in the differential expression of genes between human population groups. The authors tested in a candidate regions identified, through pre-selected mathematical models, the role of microsatellites polymorphisms in the expression of the GPX7 gene with oxidative stress. The study addresses relevant issues with direct implications for evolutionary medicine.

Major review:

The great challenge of this study is to harmonize information from different public datasets to be able to test the main hypothesis.

I would like to invite the authors to reflect and discuss the impact of using these datasets on the results obtained: (i) EBML are identified from 5 main ethnic groups (data set 1KG - AFR, EUR, SOA, EAS, AME); (ii) eSTR are identified from the GTEX database, which presents an ethnic bias, where the majority of the samples that they integrate are declared “white” (~ 85%); (iii) Data from RNAseq (GEUVADIS consortium) are only available for a 1KG subsample of Africans and Europeans.

1) Why did the authors not previously filter the EBML for AFR and EUR, since confirmatory analyses can be performed only for these groups?

2) How can the ethnic bias of GTEX samples favor the identification of EBML/eSTR EUR?

3) How many of the 313 are exclusively AFR and EUR? (maybe this info is in table S3, but it has not been made available for review)

A section with caveats would be welcome for the study.

1) Discussion of challenges to obtain reliable allele calls for microsatellites from NGS low coverage data

2) Limitations for RNAseq comparison and quantification of gene expression (mapped reads x NGS low coverage)

3) As shown in other studies, many EMBL and eSTR are found in regions with genetic signatures of natural selection. In this sense, it is important to discuss the possible hitchhiker effect on microsatellites (causal variant x tag variant).

Minor review:

1) At various points throughout the text, especially in the results section, the authors state that the partially presented results support the hypothesis that microsatellites contribute to the expression's differential. This is repetitive and is not necessary.

2) Title: the authors have confirmed data and analyses for differential expression and GPX7 only for African and European populations. For this reason, I suggest that the title of the study is not so comprehensive and is restricted to these two population groups;

3) line 147 .. “Tabulating the 313 candidate regions…”

According to figure 1d, nerve tibial genes have greater overlap for the total of the 5 super-populations. If we look only at Africans it seems to me that the thyroid is bigger. Please consider revising the sentence.

Reviewer #2: The authors have identified overlapping EBML/eSTRs and have characterized them according to ethnicity. They have found that 32 affected genes were differentially expressed in Africans and Europeans and quantified their relative contribution to differential expression. They found that repeat length correlated with expression for 15 of the 32 genes and that 2 were involved in glutathione metabolism.

Based on that, they used a mathematical model of glutathione metabolism to check how a single polymorphic microsatellite affects phenotype and proposed that microsatellite polymorphisms affect GPX7 expression and oxidative stress in Africans and Europeans.

This is a very well-conducted work. Data sounds relevant and worthy.

6. PLOS authors have the option to publish the peer review history of their article (what does this mean?). If published, this will include your full peer review and any attached files.

Reviewer #1: No

Reviewer #2: No

---

## [Author Response · Author response to Decision Letter 0]

26 Feb 2021

[our response to editor and reviewer comments are attached as a separate document]

---

## [Decision Letter · Decision Letter 1]

12 Mar 2021

Ethnically biased microsatellites contribute to differential gene expression and glutathione metabolism in Africans and Europeans

PONE-D-21-01478R1

Dear Dr. Kinney,

We’re pleased to inform you that your manuscript has been judged scientifically suitable for publication and will be formally accepted for publication once it meets all outstanding technical requirements.

Kind regards,

Danillo G Augusto

Academic Editor

PLOS ONE

Additional Editor Comments (optional):

Reviewers' comments:

Reviewer's Responses to Questions

**Comments to the Author**

1. If the authors have adequately addressed your comments raised in a previous round of review and you feel that this manuscript is now acceptable for publication, you may indicate that here to bypass the “Comments to the Author” section, enter your conflict of interest statement in the “Confidential to Editor” section, and submit your "Accept" recommendation.

Reviewer #1: All comments have been addressed

2. Is the manuscript technically sound, and do the data support the conclusions?

Reviewer #1: Yes

3. Has the statistical analysis been performed appropriately and rigorously? 

Reviewer #1: Yes

4. Have the authors made all data underlying the findings in their manuscript fully available?

Reviewer #1: Yes

5. Is the manuscript presented in an intelligible fashion and written in standard English?

Reviewer #1: Yes

6. Review Comments to the Author

Reviewer #1: This was already an informative and interesting study, and the authors have done a very nice job of responding to my comments.

7. PLOS authors have the option to publish the peer review history of their article (what does this mean?). If published, this will include your full peer review and any attached files.

Reviewer #1: No

---

## [Editor Report · Acceptance letter]

16 Mar 2021

PONE-D-21-01478R1 

Ethnically Biased Microsatellites Contribute To Differential Gene Expression and Glutathione Metabolism in Africans And Europeans 

Dear Dr. Kinney:

I'm pleased to inform you that your manuscript has been deemed suitable for publication in PLOS ONE. Congratulations! Your manuscript is now with our production department. 

Kind regards, 

on behalf of

Dr. Danillo G Augusto 

Academic Editor

PLOS ONE